# A national professional development program fills mentoring gaps for postdoctoral researchers

Ting Sun[1], Denise Drane[2], Richard McGee[3], Henry Campa, III[4], Bennett B. Goldberg[1], Sarah Chobot Hokanson [5]*

1 Department of Physics & Astronomy, Northwestern University, Evanston, Illinois, United States of America,
2 Program Evaluation Core, Northwestern University, Evanston, Illinois, United States of America,
3 Feinberg School of Medicine, Northwestern University, Chicago, Illinois, United States of America,
4 Graduate School and Department of Fisheries and Wildlife, Michigan State University East Lansing, Michigan, United States of America, 5 Office of the Provost, Boston University, Boston, Massachusetts, United States of America

* sch1@bu.edu

## Abstract

*The Postdoc Academy*: *Succeeding as a Postdoc* was designed to build postdocs' skills in career transition, career planning, collaborative research, resilience, and self-reflection. This study examined self-reported changes in five skills as learners progressed through the course. Data were collected from participants who responded to both pre- and post-surveys and engaged with the course learning activities. Results from repeated measures multivariate analysis of variance revealed that all of the self-reported perceptions of skills improved significantly upon completion of the course. Hierarchical regressions revealed that underrepresented minority learners had greater gains in their development of skills in career planning, resilience, and self-reflection. Qualitative analysis of learners' responses to learning activities found that postdocs perceived networking and mentor support as contributing factors to their skill advancement while tensions among multiple obligations and concerns of uncertainties were significant challenges to applying those skills.

## Introduction

Postdoctoral scholars (hereafter "postdocs") have long been acknowledged as important contributors to scientific and economic advancement [1]. To achieve research independence and professional readiness, postdocs need to receive effective career and professional development, which provides opportunities for them to develop a set of non-technical or transferable skills [2, 3]. Hokanson et al. found that synchronous, online development workshops were important to self-reflective and skill-building practices for graduate students and postdocs [4]. Steen et al. also found that a structured professional development program for postdocs resulted in an increased self-perception of multiple skills in participants as compared to the control counterparts [5]. A meta-analytic study demonstrated that professional development was a significant predictor of upward career mobility and was associated with higher salary, promotion and career satisfaction [6]. These results have sparked an emerging interest in professional

**Data Availability Statement:** Raw participant reflection data cannot be shared publicly because of identifiable information that could be traced back to individual course participants. Data are available

from the Boston University Institutional Data Access / Ethics Committee (irb@bu.edu) for researchers who meet the criteria for access to confidential data. However, we have made anonymized survey data available at this link: https://datadryad.org/stash/share/k24kMr0dZD-EysjI0Eg-Ymx9jNevKEtZNVf_wVfus5o.

**Funding:** SCH and BBG are multi-PIs of National Institutes of Health award 5R25GM121257-05. DD, HC, TS, and RM are all funded key personnel. https://reporter.nih.gov/search/d0cZl1uWB0uBzLXgcdilrw/project-details/10410470 The funders had no role in study design, data collection and analysis, decision to publish, or preparation of the manuscript.

**Competing interests:** The authors have declared that no competing interests exist.

development and have motivated institutions' postdoc offices and funding agencies to design and implement career and professional development programs for postdocs. A small sampling of these include: Preparing Future Faculty Program support by the Council of Graduate Schools and the American Association of Colleges and Universities; the Burroughs Wellcome Fund Career Guidance for Training award; NIH National Institute of General Medical Science, Innovative Program to Enhance Research Training grant; and, most recently, the NIH Broadening Experiences in Scientific Training (BEST) program [7], which aimed to provide career training opportunities for biomedical doctoral and postdoctoral researchers.

Massive Open Online Courses (MOOCs) offer a cost-effective, flexible and convenient adult learning option for career advancement and professional development. Authored through partnerships with leading institutions, MOOCs deliver high-quality content. A variety of MOOCs have been offered to broad audiences on platforms such as edX, or Coursera to facilitate transferable skill building or career transitioning (e.g., University Teaching 101, The Inclusive STEM Teaching Project, Career Discovery Specialization) but few target postdocs exclusively.

## Statement of the problem

Numerous challenges exist with current professional development opportunities for postdocs offered at academic institutions and/or through professional societies. Many programs focus on academic careers (e.g., The Preparing Future Faculty Program) and development for non-academic career paths can be limited [8, 9]. This, despite the fact that only 15% of postdocs secure tenure-track faculty positions [2, 10]. Second, some professional development opportunities cannot reach their target audiences because their offerings conflict with postdocs' research and personal schedules [11]. Lastly, some relatively large-scale professional development opportunities focus exclusively on biomedical disciplines largely ignoring other disciplines (e.g., BEST) [7]. We have designed, implemented, and evaluated a professional development program that offers a broad focus on skills applicable to academic and non-academic careers, flexible delivery and engagement, and covers a diversity of career opportunities for postdocs across disciplines.

Mentoring plays an essential role in supporting postdocs in their professional development engagement [12]. Yet some mentors are either unaware of professional development opportunities (see above) or may not see the benefit to their postdoc's research and career success [13]. Consequently, some postdocs are not provided with consistent and sufficient resources or opportunities for professional development. In addition, postdocs with inadequate mentoring have difficulty in balancing their research and professional development engagement [13]. The situation is even worse with underrepresented minorities (URM) and women [14, 15]. URM refers to individuals from racial and ethnic groups such as Black or African American, Hispanic or Latino, American Indian or Alaska Native, Native Hawaiian, and other Pacific Islander [16]. URM postdocs and early career faculty were found to have less mentoring as compared to their majority counterparts [17]. Additionally, due to marginalities derived from their ethnicity and gender, URM postdocs have more challenges in accessing and navigating professional development opportunities and resources [13, 18]. Developing and implementing an open access, inclusive and structured professional development program could help meet postdoc mentoring needs or supplement interpersonal and institutional mentoring, which can contribute to success for diverse postdocs.

## The postdoc academy program

The Postdoc Academy is a comprehensive professional development program for postdocs everywhere, designed to reach those in both research-intensive and well-resourced

institutions, as well as those who are isolated with limited local networking or professional development opportunities. The Postdoc Academy was motivated by the national reports that described the needs of postdocs [19], the recognition that they are an oft-ignored segment of the academic pathway, and the strong positive impact of professional development on postdoc career success [20]. The Postdoc Academy, which consists of two online asynchronous courses that also have optional in-person synchronous meetings, aims to build skills aligned with the National Postdoctoral Association core competencies [21], and supports career advancement throughout the postdoc pathway. As a blended learning program, it offers multiple learning modalities to reach diverse audiences. This program developed and implemented two Massive Open Online Courses (MOOCs) offered on the edX platform, *The Postdoc Academy*: *Succeeding as a Postdoc* and *The Postdoc Academy*: *Building Skills for a Successful Career*. This paper examines the first course that was launched, *The Postdoc Academy*: *Succeeding as a Postdoc*.

*Succeeding as a Postdoc* consists of four modules, *Finding Success as a Postdoc*, *Building an Actionable Career Plan*, *Developing Resilience*, and *Working Effectively in an Intercultural Environment*, with the aim to build learners' skills in career transition, career planning, collaborative research, resilience and self-reflection. There are various ways for learners to engage the course. Learners can participate in the course by watching course videos or reading video transcripts, participating in discussions, completing individual reflections, and engaging with interactive learning activities (Fig 1). Learners self-pace their course engagement, customizing their learning experience. The course was offered five times from January 2020 through January 2022. Each offering lasted six weeks, with one being offered before the COVID-19 pandemic and four being offered after the pandemic.

Growing evidence shows that non-technical or transferable skills are critical to postdoc success [22]. For example, research suggests a positive correlation between career transition, planning, collaborative research, resilience and self-reflection and postdoc success [6, 23–26]. Since these skills are central to the learning outcomes of *Succeeding as a Postdoc*, it is important to examine correlations in our learners associated with the development of these skills. It is also

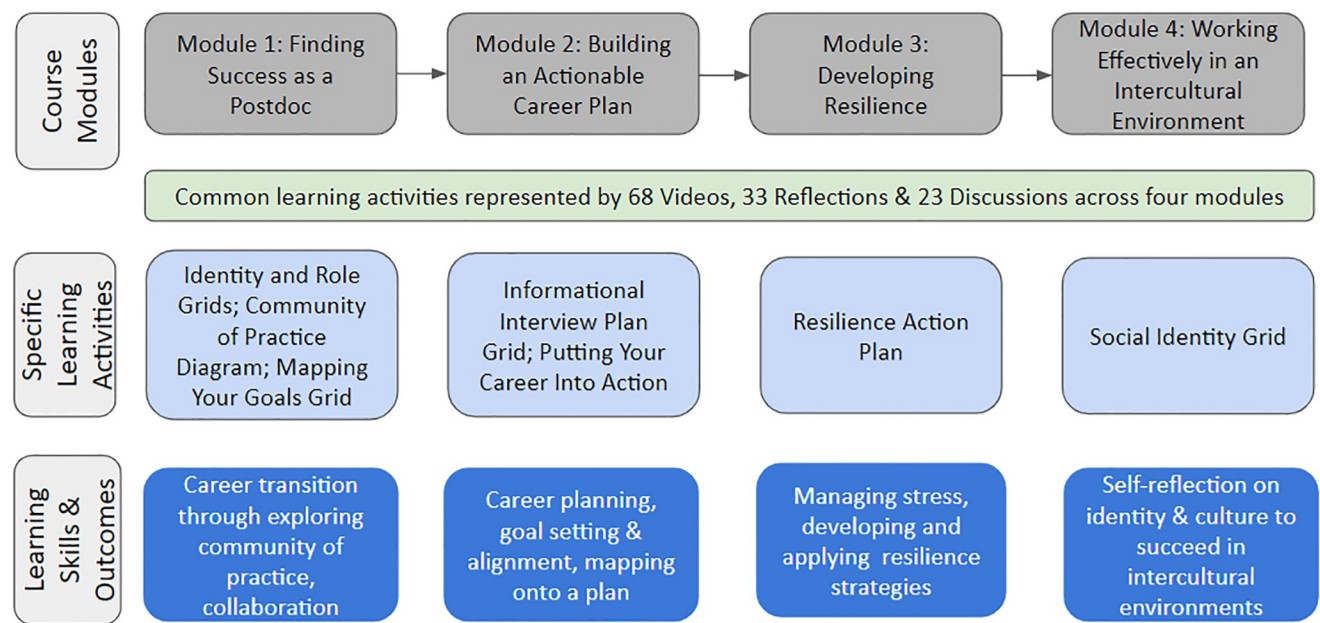

**Fig 1. Modules, learning activities, and learning skills and outcomes of *The Postdoc Academy*: *Succeeding as a Postdoc* from February 2020 through January 2022.**

important to examine how different demographic groups navigate professional development opportunities, since existing literature indicates that URM and female postdocs had unique challenges navigating professional development opportunities [13]. We sought to examine the learner perception changes in the five skills as learners progressed through the course. The research questions that guided our study are:

1. To what extent did learners' self-reported skills (career transition, career planning, collaborative research, resilience, and self-reflection) change after they participated in *The Postdoc Academy*: *Succeeding as a Postdoc*?

2. What are the differences in the changes of learners' self-reported skills as a function of learners' demographic characteristics (i.e., gender, ethnicity, discipline, and country of origin)?

3. What were learners' reflections on skill development as described in their learning experiences as they engaged with *The Postdoc Academy*: *Succeeding as a Postdoc*?

## Materials and methods

### Study design

This study used a mixed methods design with quantitative and qualitative methods. A single-group, pretest-posttest design was used to measure self-reported learning gains and behavioral intentions. Although not the strongest experimental design, the single-group design is widely used in education situations where having more than one form of instruction simultaneously is not feasible [27]. Qualitative analysis of learners' direct work products (course learning activities) was incorporated to elicit in-depth answers to the first two research questions and to enrich our understanding. A detailed description of the course learning activities can be found in the Data Sources section.

### Participants

Study participants were those who responded to both pre- and post-course surveys. The number of survey respondents, course learners, registrants, and completers for each of the *Succeeding as a Postdoc* offerings are presented in Table 1. A total of 5581 learners enrolled in *Succeeding as a Postdoc* and 473 completed the course. The total numbers of survey

**Table 1. Number of survey respondents, course learners, registrants, and completers by course run from *The Postdoc Academy*: *Succeeding as a Postdoc* (from February 2020 through January 2022).**

| Sample | Run 1 /January 2020 (RR) | Run 2 /June 2220 (RR) | Run 3 /October 2020 (RR) | Run 4 /April 2021 (RR) | Run 5 /January 2022 (RR) | Total (averaged RR) | Valid Total |
|---|---|---|---|---|---|---|---|
| Pre-Course | 396 (31%) | 442 (28%) | 153 (19%) | 193 (20%) | 150 (16%) | 1334 (23%) | 1151 |
| Post-Course | 105 (8%) | 113 (7%) | 33 (4%) | 47 (5%) | 78 (8%) | 376 (7%) | |
| Matched Sample | 62 | 91 | 17 | 35 | 33 | 238 | 215 |
| Course Learners | 560 | 686 | 309 | 323 | 334 | 2213 | |
| Registrants | 1271 | 1592 | 807 | 954 | 957 | 5581 | |
| Completers | 116 | 162 | 53 | 64 | 78 | 473 | |

*Note.* RR = Response Rate; Course learners refers to learners who interacted with at least one of the activities (e.g., watching videos, submitting reflections, posting discussions, doing learning activities) in *Succeeding as a Postdoc*.

respondents were 1334 and 376 for the pre- and post-course surveys, respectively. The current study used the sample who responded to both the pre- and post-course surveys (i.e., the matched sample). Twenty-three participants who did not provide written consent to use their survey responses for research were removed from the study, yielding 215 participants. A subset of the 215 participants ranging from 166 to 209 who interacted with *Succeeding as a Postdoc* course learning activities was also created for the learning activities analysis.

The demographic information for the sample who responded to both surveys is shown in S1 Table. The sample is predominantly postdocs ($n$ = 178, 83%), followed by doctoral students ($n$ = 13, 6%), administer or staff ($n$ = 3, 1%), faculty ($n$ = 1, 0.4%), and non-faculty researcher ($n$ = 1, 0.4%). The sample consisted of 146 women (68%). White/Caucasian participants made up 42% of the sample, followed by Asian or Asian American (19%), Hispanic or Latino/Latina/Latinx (12%) and smaller numbers of other racial/ethnic groups. Approximately half of the learners (56%) worked in biological/medical sciences, and learners working in physical/engineering/computer sciences and humanities/social sciences represented 14% and 15% of the sample, respectively. Learners from 37 countries of origin were included in the sample, with the United States accounting for the largest proportion (43%). The demographic composition of pre-course survey respondents and the national postdoc population are also presented in S1 Table to demonstrate the representativeness of the sample used in the study. The demographic composition of the sample is similar to those who responded to the pre-course survey. However, our sample had more women, more URM learners, and learners working in humanities/social sciences compared with the national postdoc survey respondents [28].

## Data sources

**Surveys.**   Survey instruments used in the current study include pre- and post-course surveys. Surveys were designed by embedded evaluators and course designers based on the theory of change for each of the four course modules, which connected course pedagogical design, learning activities and short-term outcomes [29]. The pre-course survey had five items asking participants to rate (using a five-point Likert scale; 1 = Not proficient to 5 = Extremely proficient) their self-perceptions of their competency in making career transitions skillfully, planning career, developing collaborative research relationships, rebounding from setbacks or challenges, and engaging in self-reflection, which are central to the learning goals and theory of change of the modules in *Succeeding as a Postdoc*. The pre-course survey also had items collecting participants' demographic information including gender, ethnicity, discipline, and country of origin. The post-course survey included the same five items assessing participants' perceptions of their skills upon completion of *Succeeding as a Postdoc*. The post-course survey also had items evaluating the quality of each module, which were not used in this study. The internal consistencies (Cronbach's alpha) of the five items were 0.73 and 0.81 for the pre- and post-course surveys, respectively, indicating good internal consistencies of participants' responses to the surveys.

**Course learning activities.**   Eight course learning activities (*Identity Grid*, *Role Grid*, *Mapping Your Goals Grid*, *Informational Interview Plan Grid*, *Putting Your Career Into Action*, *Social Identity Grid*, *Community of Practice Diagram*, *and Resilience Action Plan*) were developed as part of course content to facilitate building deeper knowledge through application of module concepts (S2 Table). The grid-type activities consisted of multiple levels of prompts to which learners responded, each building upon the last and progressing toward a goal, with video guidance and examples interspersed between stages (an example of *Putting Your Career Into Action* is included in S1 Text). The pedagogical design of these activities engaged learners in thinking through progressive steps to construct for themselves a process to advance and

apply the skills they were learning. Learners needed to complete at least five activities to achieve course completion and receive a certificate. We chose six prompts (1. Obstacles on your pathway to success; 2. What I'm currently not doing 3. Why I'm currently not doing it 4. Strategies to overcome obstacles; 5. What will help you to achieve your goal? 6. What are the challenges/barriers you might encounter?) with detailed open responses from *Mapping Your Goals* and *Putting Your Career Into Action* from the eight course learning activities for qualitative analysis. The six prompts were chosen because they are open-ended questions and provided in-depth responses regarding learners' awareness, knowledge, and ability to apply concepts than the single word or phrase entries in, for example, the *Identity Grid activity* (additional analysis of this and other activities will be the subject of a forthcoming manuscript, since they yield findings along different themes). We analyzed the responses as direct work products that speak to learners' state of mind at that stage of the course and their own professional development.

## Data collection

The Institutional Review Board (IRB) of the researchers' universities approved the study (IRB approval numbers is 5419X) and the participants who agreed to allow the information collected to be used for research were included in the study. Both surveys were embedded in the *Succeeding as a Postdoc* course on the edX platform and data were collected through Qualtrics. The pre-survey was administered during the first week of the course and the post-survey was administered in the sixth week of the course. Participants completed each survey in approximately 15 minutes. Course learning activities were embedded in *Succeeding as a Postdoc* and learners' responses were collected and managed through PostgreSQL database as they progressed through each offering of the course.

## Data analysis

Data preparation and cleaning involved handling duplicates and missing values. If two response instances of the same individual were at the same level of completeness, the most recent one was retained. The percentage of missing values on the outcome variables ranged from 2% to 10% and that on predictor variables ranged from 8% to 11%. Little's test [30] was first conducted using the R function *mcar_test* and the result revealed that the data were missing completely at random (MCAR). Multiple imputations were used to account for missingness using $m$ = 5. Multiple imputation has been suggested to be advantageous over other alternative techniques (e.g., pairwise deletion, listwise deletion, single imputation) in handling missing values [31]. Prior to data analyses, we ensured that the assumptions for the repeated measures multivariate analysis of variance (MANOVA) and regressions had been met, which include independence, multivariate normality of dependent variables, linearity, homoscedasticity, and absence of multicollinearity [32].

A repeated measures MANOVA was conducted to determine if there were significant differences in participants' self-perceptions of skills between the pre- and post-course surveys, our first research question. MANOVA was chosen instead of multiple ANOVAs or t-tests, because multiple tests would have increased the risk of committing Type I errors [32]. To answer the second research question, hierarchical linear regressions were conducted with gender, ethnicity, discipline, and country of origin entered as predictors and skills at the post-course survey entered as outcome variables. Skills in the pre-course survey were entered in the first step as the covariate. All the predictors were dummy coded. Ethnicity was coded as 1 for majority (i.e., White/Caucasian and Asian or Asian American) and 0 for URM (i.e., Alaska Native or Native American, Black or African American, Hispanic or Latino/Latina/Latinx,

Middle Eastern or Northern African, Pacific Islander, and Multiracial groups). Gender was coded 1 for women and 0 for men, since we did not have sufficient numbers of non-binary participants. For discipline data, two dummy variables, discipline (biological/medical) and discipline (physical) were created with 0 being the reference variable and referring to the discipline of humanities/ social sciences, and 1 referring to biological/medical sciences and physical/engineering/computer sciences, respectively. Based on the distribution of countries by Human Development Index (HDI) [33], the variable of Country was dummy coded into the variable of country (highHDI), with 1 referring to very high and high HDI countries, and 0 referring to the medium and low HDI countries. The regression model is:

$$post_{ij} = \beta_{0j} + \beta_{1j}pre_{ij} + \beta_{2j}female_{ij} + \beta_{3j}majority_{ij} + \beta_{4j}Discipline(biological/medical)_{ij}$$
$$+\beta_{5j}Discipline(physical)_{ij} + \beta_{6j}Country(highHDI)_{ij} + r_{ij}$$

where $j$ refers to each of the five skills (career transition, career planning, collaborative research, resilience, and self-reflection).

For the third research question regarding learners' reflections on skill development, we conducted a thematic analysis [34] to examine learners' responses to the six prompts of course learning activities. Coding occurred in two intentional phases. In the first phase of analysis, one researcher read all the open-ended responses. Open coding was adopted to locate and identify codes based on each prompt. After the code identification and exploration phase, coded information was reread to identify underlying connections between them and codes were categorized into themes and patterns. Several approaches were adopted to ensure the quality of data analysis and trustworthiness of findings. First, identified codes and themes were refined iteratively through dynamic discussions with team members. Second, 15% of the data was re-coded after three weeks to check diachronic reliability [35]. We obtained a reliability coefficient of 0.91, indicating the consistency and stability of the data analysis and interpretation. It is also important to note that the study team member who did the primary coding is a postdoc. The intersecting identity between the researcher and the participants contributed to the understanding of data and interpretation of findings. The remaining team members who contributed through discussion and alternative interpretations have extensive experience with research training at pre- and postdoc levels, and qualitative research methods.

## Results

### Self-perceptions of skill change

Results from the repeated measures MANOVA revealed that the omnibus Wilks's lambda was statistically significant for the main effect of time, indicating that the combined dependent variables differed, on average, between pre-course and post-course surveys, Wilk's = .623, $F(5, 209) = 25.28$, $P < 0.001$. From the univariate $F$ analysis (Table 2), there were statistically significant increases over time for each measure. Learners self-reported skills in career transition, $F(1, 213) = 25.23$, $P < 0.001$, generalized $\eta^2 = .07$, career planning, $F(1, 213) = 44.97$, $P < 0.001$, generalized $\eta^2 = .12$, collaborative research, $F(1, 213) = 18.46$, $P < 0.001$, generalized $\eta^2 = .05$, resilience, $F(1, 213) = 29.04$, $P < 0.001$, generalized $\eta^2 = .09$, and self-reflection, $F(1, 213) = 21.25$, $P < 0.001$, generalized $\eta^2 = .07$, all improved significantly after they took *Succeeding as a Postdoc*. Effect size is interpreted as small when $\eta^2 = 0.01$, medium when $\eta^2 = 0.06$, and large when $\eta^2 = 0.14$ [36]. According to Cohen's [36] benchmark, all the effect sizes ranged from medium to large, with the effect on career planning and resilience being the largest and the effect on collaborative research being the smallest.

**Table 2. Means and standard deviations of self-reported skills on the pre- and the post-course surveys, and the follow-up repeated measures ANOVA results.** Data were from *The Postdoc Academy*: *Succeeding as a Postdoc* from February 2020 through January 2022 (pre-course survey, $n = 214$, post-course survey, $n = 214$).

| Variable | Pre-Course Survey ($n = 214$) | | Post-Course Survey ($n = 214$) | | SS | df | F | $\eta^2$ |
|---|---|---|---|---|---|---|---|---|
| | M | SD | M | SD | | | | |
| career transition | 2.8 | 0.9 | 3.3 | 0.8 | 25.23 | 1 | 58.25*** | 0.07 |
| career planning | 2.9 | 1.0 | 3.5 | 0.9 | 44.97 | 1 | 89.26*** | 0.12 |
| collaborative research | 3.1 | 0.9 | 3.5 | 1.0 | 18.46 | 1 | 37.85*** | 0.05 |
| resilience | 3.3 | 0.9 | 3.8 | 0.8 | 29.04 | 1 | 70.89*** | 0.09 |
| self-reflection | 3.5 | 1.0 | 4.0 | 0.9 | 21.25 | 1 | 40.51*** | 0.07 |

*Note.* SS = Sum of Squares, *df* = degree of freedom;

***$P < 0.001$, two-tailed.

## Differences of skill changes by ethnicity, gender, discipline and country

Results from hierarchical linear regressions revealed that ethnicity had a significant influence on career planning ($t = -2.23$, $P < 0.05$), resilience ($t = -3.87$, $P < 0.001$), and self-reflection ($t = -2.09$, $P < 0.05$) after controlling for differences in baseline skills on the pre-course survey. The unstandardized regression coefficient (B) indicated that URM learners reported 0.30, 0.47, 0.24 greater in their self-perceptions of career planning, resilience, and self-reflection, respectively, as compared to their majority counterparts. This result suggested that URM learners had greater gains in their self-perceptions of skills in the three aspects. However, no statistically significant differences were identified in perceived skill development as a function of gender, discipline, and country of origin (Table 3). In other words, although learners perceived that their skills improved after taking *Succeeding as a Postdoc*, the amount of skill development was not different between men and women, and among people from different disciplines and country backgrounds.

For this section and those that follow, we use "postdoc" to associate our findings with the large majority of learners since we demonstrate that no differences occur in the direction and significance of the estimates between the total data set and the sub-set that is only postdocs (see S3 and S4 Tables for the results with the sub-set of only postdocs).

## Reflections on learning from direct work product

Three themes were identified and the top (most frequent) four codes for each prompt between majority and URM learners were also compared (Fig 2) because significant differences in skill changes were found as a function of ethnicity. The top four codes were very similar, although the frequency order among the four differed between URM and majority postdocs.

**Networking.** One of the most salient reflections among postdocs was that they perceived networking as necessary components for their career success. "Not networking" was one of the strong and consistent codes that emerged from participants' responses to the prompt "What are you not currently doing". Approximately half of the participants (45% of URM, $n = 58$, and 50% of majority, $n = 119$) mentioned that they neither "build their professional network with potential collaborators" nor "proactively reach out to other postdocs".

> "Outside of people I directly work/collaborate with for my research, my network is basically non-existent. I don't do a good job at conferences introducing myself to people in my field and actually laying the foundation for my network".

**Table 3. Pooled hierarchical regressions for gender, ethnicity, discipline and country of origin predicting skill development.** Data were from *The Postdoc Academy*: *Succeeding as a Postdoc* from February 2020 through January 2022 ($n$ = 214).

| | Career Transition | | | Career Planning | | | Collaborative Research | | | Resilience | | | Self-Reflection | | |
|---|---|---|---|---|---|---|---|---|---|---|---|---|---|---|---|
| | *B* | *SE(B)* | *t* | *B* | *SE(B)* | *t* | *B* | *SE(B)* | *t* | *B* | *SE(B)* | *t* | *B* | *SE(B)* | *t* |
| **Step 1** | | | | | | | | | | | | | | | |
| pre-survey | 0.41 | 0.06 | 6.61*** | 0.36 | 0.07 | 5.54*** | 0.40 | 0.07 | 7.36*** | 0.42 | 0.06 | 6.60*** | 0.32 | 0.06 | 5.58*** |
| $R^2$ | 0.19*** | | | 0.16*** | | | 0.20*** | | | 0.18*** | | | 0.13*** | | |
| **Step 2** | | | | | | | | | | | | | | | |
| pre-survey | 0.40 | 0.06 | 6.37*** | 0.35 | 0.07 | 5.23*** | 0.51 | 0.07 | 7.40*** | 0.41 | 0.06 | 6.56*** | 0.33 | 0.06 | 5.78*** |
| woman | 0.18 | 0.13 | 1.45 | 0.08 | 0.14 | 0.55 | 0.13 | 0.14 | 0.90 | -0.21 | 0.14 | -1.47 | 0.17 | 0.13 | 1.36 |
| majority | -0.15 | 0.13 | -1.20 | -0.30 | 0.13 | -2.23* | -0.17 | 0.14 | -1.25 | -0.47 | 0.12 | -3.87*** | -0.24 | 0.11 | -2.09* |
| Discipline(biomedical) | 0.06 | 0.14 | 0.42 | -0.05 | 0.15 | -0.33 | -0.10 | 0.15 | -0.68 | -0.03 | 0.12 | -0.23 | -0.08 | 0.13 | -0.64 |
| Discipline(physical) | -0.14 | 0.16 | -0.83 | -0.12 | 0.18 | -0.69 | -0.34 | 0.18 | -1.86 | 0.01 | 0.15 | 0.08 | -0.07 | 0.16 | -0.45 |
| Country(highHDI) | -0.18 | 0.17 | -1.05 | -0.09 | 0.18 | -0.48 | 0.09 | 0.21 | 0.42 | 0.14 | 0.16 | 0.91 | -0.10 | 0.14 | -0.67 |
| $R^2(\Delta R^2)$ | 0.22(0.03) | | | 0.20(0.04) | | | 0.24(0.04) | | | 0.26(0.08)** | | | 0.16(0.03) | | |

*Note.*

*P < 0.05,

***P < 0.001, two-tailed;

Ethnicity was coded as 1 for majority (i.e., White/Caucasian and Asian or Asian American) and 0 for URM (i.e., Alaska Native or Native American, Black or African American, Hispanic or Latino/Latina/Latinx, Middle Eastern or Northern African, Pacific Islander, and Multiracial groups). Gender was coded as 1 for women and 0 for men. For discipline data, two dummy variables, discipline (biological/medical) and discipline(physical) were created with 0 being the referent variable and referring to the discipline of humanities/social sciences, and 1 referring to biological/medical sciences and physical/engineering/computer sciences, respectively. Country was dummy coded into the variable of country (highHDI), with 1 referring to very high and high HDI countries, and 0 referring to the medium and low HDI countries.

However, postdocs realized that networking was a contributing factor to the success of their career as they engaged with the course, and it was among the top strategies that learners said they would use to overcome obstacles on their pathway to success. One postdoc made a plan to seek networking opportunities, saying:

> *"Over the next week, [I] will actively seek out and connect with individuals in my network who are professionals in research management or who have connections in the profession."*

Participants noted that networking opportunities are necessary for successful career transition and planning and play a vital role in establishing research collaborations. One postdoc planned to improve networking by engaging with PIs and attending conferences and believed that s/he would build "possible collaboration with other colleagues". In addition, connecting to "people who have gone through similar experiences" would help them "relieve stress, build confidence and develop resilience".

**Perspectives on mentor support.** Around one quarter of postdocs (27% of URM, $n$ = 59, and 23% of majority, $n$ = 122) perceived a lack of mentor support as a (potential) barrier to the achievement of their career goal, as evidenced in their responses to the prompt "What are the challenges/barriers you might encounter". Despite that awareness, participants stated that they were not "proactively seeking out mentors" or "talking to existing mentors" due to the devotion of their time to other commitments such as research.

Postdocs reflected on the importance of mentorship while engaging with the course. Mentor support was the most salient theme emerging from their responses to the prompt "What will help you to achieve the goal" and a majority of learners believed that consistent and effective mentorship would bring intellectual or social benefits to their career success by offering

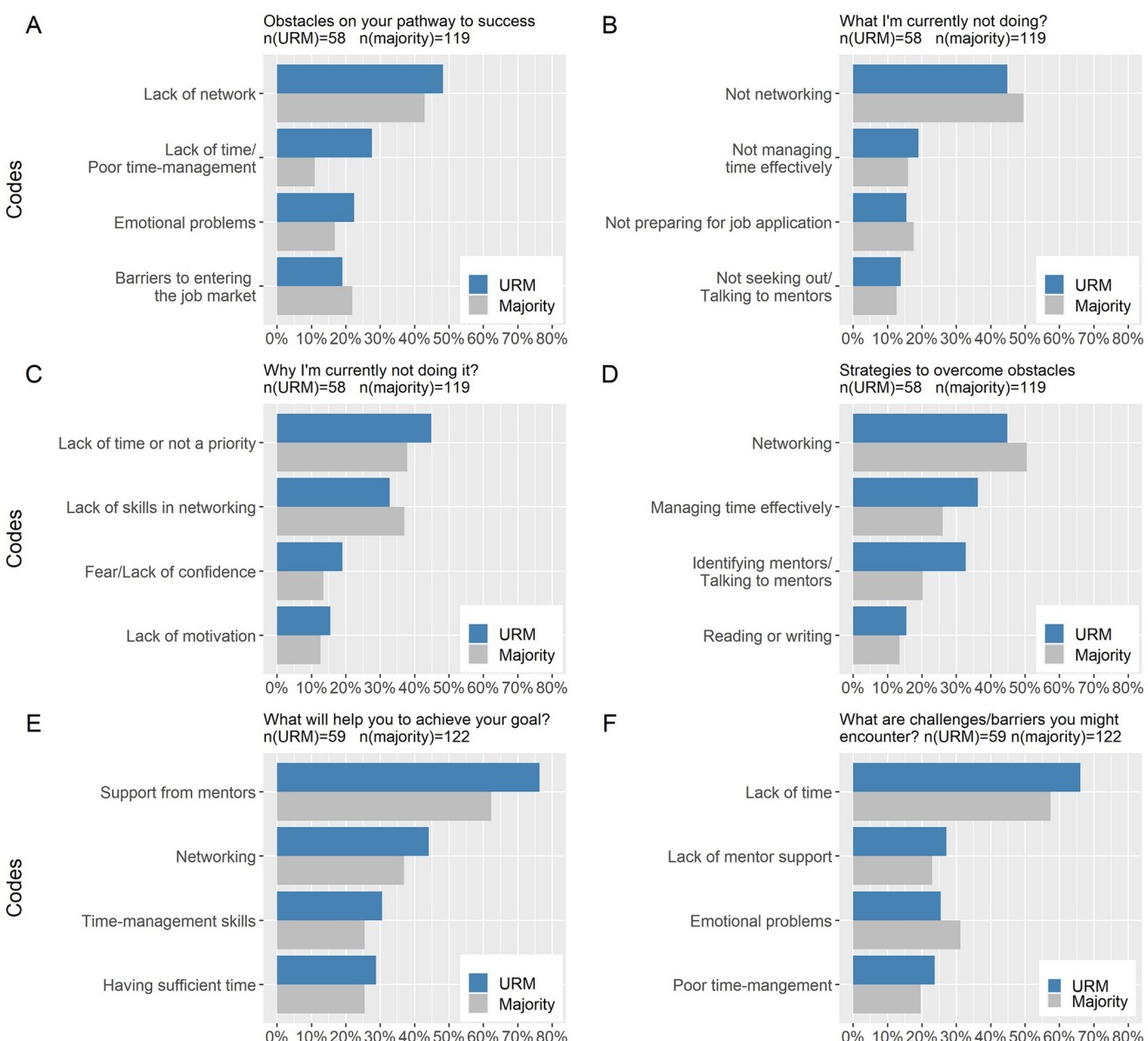

**Fig 2. The top four most frequent codes of each prompt for majority and URM learners.** Data were from responses to the six open-ended prompts of learning activities (see above) from *The Postdoc Academy*: *Succeeding as a Postdoc* from February 2020 through January 2022.

"feedback and guidance" in writing manuscripts and grant proposals and providing "network opportunities and accessibility to resources". This finding is prominent for URM postdocs, which was endorsed by 76% (*n* = 59) of URM postdocs as compared to 62% (*n* = 122) of majority postdocs. Thus, postdocs noted that they would either "identify a mentor" or "ask current mentors" for connections or information. One URM postdoc stated:

> "[I would] [a]sk mentors for names of people to connect with or to actually connect me with them, try to start with someone I already know, talk to mentors about doing informational interviews so that I'm more likely to do it if I've talked to them about it."

**Challenges in engendering changes.** Challenges are inevitable in any effort to engender and support change. Challenges that are salient in postdocs' responses are a tension between a job and a career and fear of uncertainties.

*The tension between fulfilling current job requirements and exploring a future career.* Postdocs comprise a chronically overworked population, who enter needing to complete prior research, fulfill the obligations of current jobs, and prepare for their future careers [37]. A tension between a current job and a future career is a strong and consistent theme in our learners' responses. Insufficient time and demanding workload were perceived as the main sources of their resistance to engaging in career preparation such as career-planning or collaboration-building behaviors, including conducting informational interviews, establishing social connections or seeking professional development opportunities. This theme is also more salient for URM learners. For example, over a quarter (28%, $n = 58$) URM postdocs perceived lack of time or time-management skills as a major obstacle on their way to success compared to only 11% ($n = 119$) for majority learners. Likewise, a majority of minority URM (66%, $n = 59$) believed that lack of time was the biggest challenge or barrier they might encounter in realizing their goals, which was endorsed by 57% ($n = 122$) of majority postdocs. In addition, 45% ($n = 58$) of URM and 38% ($n = 119$) of majority postdocs perceived "lack of time or not a priority" as the major reason that they were not seeking professional development opportunities or establishing networking. One postdoc commented:

> *"I've been heavily side-tracked with work and personal stuff that I often forget to contact others that I've met at conferences to network."*

Faced with competing obligations, postdocs tended to prioritize their time to work on research-related tasks such as experiments, manuscripts and grants. One learner expressed this perspective by stating:

> *"Lack of time or better use of the time. I have been focused on writing manuscripts and performing experiments. . .So, the real challenge is considering this task as 'minor'."*

*Fear of uncertainties.* Learning activity responses revealed that postdocs were experiencing a multitude of negative emotions: imposter phenomenon, anxiety, stress, and low self-efficacy, which functioned as significant barriers to their proactive "help-seeking" or "reaching out" behaviors. Some also claimed that it was their "shy and intimidated" disposition that explained their hesitance to change. These negative emotions, coupled with the introverted disposition, induced a sense of uncertainty and fear, which would hold them back in networking or seeking collaboration opportunities. One learner listed their fears and hesitations:

> *"I'm afraid of being rejected when trying to interact with people. Also, I'm afraid of not understand[ing] when talking to people in English because sometimes they speak really fast. Additionally, I'm afraid of not being knowledgeable when talking to peers in conferences and I don't want to look like a fool or stupid. And on top of all this I HATE SMALL TALK and I feel that is what I need to start conversations with new people."*

Additionally, we compared and displayed the four codes by gender, discipline and country (S1–S3 Figs). Only one consistent difference was noted by gender across the prompts. The theme of networking was more prominent for women postdocs. For example, 41% of women postdocs perceived lack of skills in networking as a major challenge to career preparation, social connections, or professional development. In contrast, only 19% of male postdocs

reported the same challenge. Other than this, no clear patterns were identified in the comparison of codes across different groups.

## Discussion

We examined the changes of postdocs' self-reported skills in career planning, transition, collaborative research, resilience and self-reflection as they engaged with a professional development course, *Postdoc Academy*: *Succeeding as a Postdoc*. We found that postdocs' perceptions of their skills improved significantly upon completion of the course. Also, URM postdocs had greater gains in their self-perceived career planning, collaborative research and resilience, relative to majority postdocs. Qualitative analysis of course learning activities found that postdocs perceived networking and mentorship as support of their career progression, and revealed a tension between a current job and a future career and fear of uncertainties as significant barriers to improvement.

All five self-reported skills improved significantly, providing evidence of the effectiveness of *Postdoc Academy*: *Succeeding as a Postdoc* in supporting postdocs' skill development in career transition, career planning, collaborative research, resilience and self-reflection, which are necessary for their success in and beyond a postdoc appointment. We hypothesize that the growth might be attributed to the inclusive and active-learning pedagogy, which created an accessible professional development environment for postdocs. Also, we believe videos that share stories and perspectives of diverse postdocs invite diverse learners to engage. We know from previous work that integrating self-reflection prompts and interactive activities create opportunities for participants to apply module concepts and directly facilitate their learning [4]. Notably, the course had larger effect sizes on the advancement of skills of career planning and resilience whereas had relatively smaller effect sizes on the growth of the skill of collaborative research, entirely consistent with the fewer number and depth of activities and course content related to collaborative research.

We found URM postdocs had greater gains in their self-reported development of skills in career planning, resilience and self-reflection as compared to their majority peers, suggesting that this course brings greater benefits to URM postdocs relative to majority postdocs. One explanation of this finding is that URM postdocs had lower level skills in the three aspects prior to entering the course, which might be caused by insufficient mentoring they received [17]. This provides support that our online, asynchronous course, *Succeeding as a Postdoc* can fill perceived needs in professional development and mentoring and is particularly beneficial for some URM postdocs.

We found postdocs perceived networking as a key contributing factor to their skill advancement in preparation for job applications, identification of career and professional opportunities, and establishment of research collaboration and resilience. Personal and professional networks are vital for postdocs because many experience a sense of isolation and detachment, which are exacerbated by remote working opportunities and the COVID-19 pandemic [38]. This is more prominent for female postdocs. This finding has been confirmed by the existing literature, which suggests that it is imperative for postdocs to establish a communication plan to expand their network or maintain connections with their local and disciplinary community [39].

We found that postdocs perceived mentor support as a significant factor influencing their career planning, transition, and collaborative research, and this perception was stronger for URM postdocs. This finding is consistent with previous research, which found that perceived mentor support during the postdoctoral period was a significant predictor of postdoc career plans [28], the odds of securing a permanent position, and future academic success [40]. McConnell et al. [28] further noted that mentor support was particularly beneficial for women

and URM postdocs in their "pursuit of research-intensive academic careers"(p. 9). Our course helps postdocs heighten the awareness of broadening their mentoring network to meet multi-faceted needs for their future success.

Our analysis found that the most significant challenges to postdocs' applying their skills to advance career development are tensions among multiple current obligations and concern of future uncertainties. URM postdocs struggled with balancing multiple responsibilities and expressed a lack of confidence generally in skills necessary to achieve their careers, likely due to imposter phenomena and lack of belonging and identity [15, 41]. Hence *Succeeding as a Postdoc* provides some of the necessary professional development learning and practice to build skills and confidence, especially important for URM postdocs. We also found that post-docs were experiencing multiple negative emotions such as stress, anxiety, sadness and worry, which may be due to the nature of their positions (impermanence of their employment, heavy workload of current jobs, and insecurities of future career) and also the impact of the COVID-19 pandemic [42]. These negative emotions produced a sense of uncertainty, which were per-ceived as barriers to their cognitive or behavioral changes. Handled inappropriately, these emotions can lead to severe mental health issues and have a negative impact on academic achievement and professional success [23].

## Limitations and future research

The current study is subject to limitations. First among them is that our participants com-prised a self-selected and engaged subgroup of postdocs who completed the course learning activities and both the pre- and post-course surveys, which may pose an issue for generalizabil-ity and transferability of our findings. We do note, however, that we observed a high degree of homophily between the demographic distribution of our sample and the national postdoc pop-ulation, indicating at least that such self-selection doesn't shift the demographic representa-tion. Second, the quantitative part of the study used a single-group pretest-posttest design, which may have threats to internal validity, including maturation and statistical regression (i.e., regression toward the mean). Third, the skills we measured were learners' self-reported skills rather than their actual skills. Lastly, we found that URM postdocs tended to have more gains in their self-perceptions of skill development, but we can only use prior studies and spec-ulation as to why. Focus groups and interviews will be conducted in the future to obtain quali-tative data to make the study more explanatory.

## Supporting information

**S1 Table. Demographic information of survey respondents and national postdoc popula-tion.**
(PDF)

**S2 Table. Brief description of the learning activities in *Succeeding as a Postdoc*.**
(PDF)

**S3 Table. Means and standard deviations of self-reported skills on the pre- and the post-course surveys, and the follow-up repeated measures ANOVA results with the sample of postdocs only.**
(PDF)

**S4 Table. Pooled hierarchical regression analysis summary for gender, ethnicity, discipline and country of origin predicting skill development with the sample of postdocs only.**
(PDF)

**S1 Text. Description of *putting your career into action* learning activity.**
(PDF)

**S1 Fig. Codes of each prompt for male and female learners.**
(TIF)

**S2 Fig. Codes of each prompt for learners in humanities/social sciences, and biological/medical sciences and physical/engineering/computer sciences.**
(TIF)

**S3 Fig. Codes of each prompt for learner in low and high HDI countries.**
(TIF)

## Acknowledgments

We thank the following individuals for their expertise and assistance throughout all aspects of our study and for their help in improving the manuscript: Anne-Sophie Bohrer, Olivia Chesniak, Noah Green, Robin Greenler, Jessica Maher, Antonio Nunez, and Celine Young.

## Author Contributions

**Conceptualization:** Denise Drane, Richard McGee, Bennett B. Goldberg, Sarah Chobot Hokanson.

**Data curation:** Ting Sun.

**Formal analysis:** Ting Sun.

**Funding acquisition:** Denise Drane, Richard McGee, Henry Campa, III, Bennett B. Goldberg, Sarah Chobot Hokanson.

**Investigation:** Ting Sun, Denise Drane, Henry Campa, III, Bennett B. Goldberg, Sarah Chobot Hokanson.

**Methodology:** Ting Sun, Denise Drane, Richard McGee, Henry Campa, III, Bennett B. Goldberg, Sarah Chobot Hokanson.

**Project administration:** Bennett B. Goldberg, Sarah Chobot Hokanson.

**Resources:** Sarah Chobot Hokanson.

**Supervision:** Denise Drane, Richard McGee, Bennett B. Goldberg, Sarah Chobot Hokanson.

**Validation:** Ting Sun.

**Visualization:** Ting Sun.

**Writing – original draft:** Ting Sun, Denise Drane, Richard McGee, Henry Campa, III, Bennett B. Goldberg, Sarah Chobot Hokanson.

**Writing – review & editing:** Ting Sun, Denise Drane, Richard McGee, Henry Campa, III, Bennett B. Goldberg, Sarah Chobot Hokanson.

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
