## [Decision Letter · Decision Letter 0]

27 Jan 2023

PONE-D-22-26426A national professional development program fills mentoring gaps for postdoctoral researchersPLOS ONE

Dear Dr. Hokanson,

Thank you for submitting your manuscript to PLOS ONE. After careful consideration, we feel that it has merit but does not fully meet PLOS ONE’s publication criteria as it currently stands. Therefore, we invite you to submit a revised version of the manuscript that addresses the points raised during the review process.

Generally this is a well written manuscript that needs some minor revisions.  As both reviewers suggest, the introduction should be have a stronger focus on published work in post-doc professional development as well as MOOCs in professional development. This will mostly likely increase the length of the introduction, but will situate the study in a broader context. Expanding the description of engage learners and how they engaged with the material would benefit this study.  Expanding the analysis of the qualitative data as suggested by reviewer 2 would enhance the quantitative results. The reviewers have several other points that should be addressed.  In addition to their comments, it is not evident in the manuscript if the definition of URM includes the international post-docs in the URM sample or only the United States URM post-docs. It would help figure 2 if each panel in the figure was given a letter and then in the manuscript when referring to panels in figure 2 use both figure 2 and the letter. That would reduce the cognitive load on readers. I also noticed a typo in Figure 1, in the bottom right box cultural is misspelled.

We look forward to receiving your revised manuscript.

Kind regards,

Sue Ellen DeChenne-Peters, Ph.D.

Academic Editor

PLOS ONE

and https://journals.plos.org/plosone/s/file?id=ba62/PLOSOne_formatting_sample_title_authors_affiliations.pdf.

2. Please change "female” or "male" to "woman” or "man" as appropriate, when used as a noun (see for instance https://apastyle.apa.org/style-grammar-guidelines/bias-free-language/gender).

* Please change "caucasian” to "white” or "European" as appropriate (see for instance https://apastyle.apa.org/style-grammar-guidelines/bias-free-language/racial-ethnic-minorities).

* Please provide additional details regarding ethical approval in the body of your manuscript. In the Methods section, please ensure that you have specified the name of the IRB/ethics committee that approved your study.

* Please provide additional details regarding participant consent. In the Methods section, please ensure that you have specified (1) whether consent was informed and (2) what type you obtained (for instance, written or verbal). If your study included minors, state whether you obtained consent from parents or guardians. If the need for consent was waived by the ethics committee, please include this information.

Reviewers' comments:

Reviewer's Responses to Questions

**Comments to the Author**

1. Is the manuscript technically sound, and do the data support the conclusions?

Reviewer #1: Yes

Reviewer #2: Yes

2. Has the statistical analysis been performed appropriately and rigorously? 

Reviewer #1: Yes

Reviewer #2: Yes

3. Have the authors made all data underlying the findings in their manuscript fully available?

Reviewer #1: Yes

Reviewer #2: Yes

4. Is the manuscript presented in an intelligible fashion and written in standard English?

Reviewer #1: Yes

Reviewer #2: Yes

5. Review Comments to the Author

Reviewer #1: This manuscript presents on an evaluation of an online course for postdoctoral fellows. The team build an online module to help postdocs with their career development. The evaluation included a pre and post survey that assessed one’s self reported perception of skill improvement. Overall, the evaluation and methodology used is rigorous, as is the data analysis. They team used both qualitative and quantitative approach for this mixed method design. The data analysis included subgroup comparisons to determine if there was a different impact for those who are underrepresented, as well as gender, country, and discipline. Some minor comments may help improve the manuscript. Overall, the manuscript is extremely well-written.

1. The introduction seems quite lengthy. This could be shortened with a sharper focus on postdoctoral training.

2. On page 8, when discussing the sample, this seems like an extremely biased sample. That is, those who completed the course and the post survey would more likely be those who saw the potential for the module to help them. Those who dropped or didn’t complete are likely those who didn’t see the value or did not think the course was helping them.

3. The definition of Engaged Learner seems quite broad. Is it that it includes anyone who has done one exercise within the module? So, if someone only watched a 5 min video, they would be considered Engaged? To me, I would see someone who is engaged as someone who has completed about 50% of the work.

4. In Table 1, Matched sample is listed as a row, but not described in text.

5. On page 11 in the discussion of the prompts, I think the explanation of the prompts could be a bit clearer. I was uncertain as to how they were presented to the postdoc and if they were part of the module’s learning content or if this was solely for qualitative analysis.

6. For those postdocs who were not on the binary, how were their data treated? Were they dropped for analysis?

7. I know the qualitative results show some trends that URM made more progress than the majority postdocs. Are you able to test for significant improvements?

Reviewer #2: This manuscript describes a massive open online course (MOOC) designed to build postdocs’ skills in career transition, career planning, collaborative research, resilience and self-reflection. Quantitative, self-reported learning outcomes data gathered from participants show skills gains in all the targeted areas of learning, with higher gains reported by historically underrepresented participants. Qualitative analysis of work products generated by course participants indicate that networking and mentor support may be crucial to advancing skills in these areas.

The premise of this study is thought-provoking, and the results are compelling. I have included some suggestions below to make the article even stronger.

While well-written overall, the article would be strengthened by the inclusion of some additional references and discussion of relevant articles, as space allows. Specifically:

• The authors could incorporate more published work on post-doc professional development, and the targeted domains of learning that have been identified and pursued, including work from scholars such as Fuhrmann and Sinche. It was especially odd that the work of Steen et. al was not included since it was referenced in the cover letter as a research article this study was built upon.

• The authors could include some additional references about the success of using MOOCs as an approach to professional development in general, or specifically for post-doc professional development.

The materials and methods section, data sources, and data analysis sections are well done. This reviewer would suggest a few additions in these sections:

• Additional clarification of when and how the participant responses to the six selected prompts are captured. Are these responses collected at a single time point at the end of the course or from a reflection activity which participants can add to over time?

• It is noted that 83% of the participants were post-docs; it would help if the authors added information on the career stages of the other 17%, if known.

• Additional clarification on how the learners in this study engaged in the various elements and modes of the course. The text notes: “Learners can participate in the course by watching course videos or reading video transcripts, participating in discussions, completing individual reflections, and engaging with interactive learning activities.” It also notes that learners are considered to have completed the course “if they do at least 5 activities.” Are the completed engagement modes and activities tracked? If so, it would be interesting to know if there were any correlations seen between skill gains and levels/types of engagements.

The quantitative results included in the article are strong and engaging. This reviewer suggests including one additional piece of quantitative data:

• Satisfaction/Usefulness of the four modules and the various approaches to learning. It is noted in the discussion: “Also, we believe videos that share stories and perspectives of diverse postdocs invite diverse learners to engage. We know from previous work that integrating self-reflection prompts and interactive activities create opportunities for participants to apply module concepts and directly facilitate their learning.” This discussion point could be supported more strongly with satisfaction data from learners across the content and modalities.

The qualitative results included in the article are intriguing and could be expanded and deepened. One possibility includes:

• Analyses of the qualitative data by gender and nationality across at least the 4 identified codes. These analyses would strengthen the link between the quantitative and quantitative results. For example, if differences were seen only for URM/non URM learners in both the quantitative and qualitative data, it would enhance understanding. If instead, the only differences seen are in the qualitative data for demographics beyond ethnicity, that would also enhance understanding. Moreover, several points in the discussion related to the qualitative data raise questions about other identities. For example, the literature exploring gender, balancing multiple responsibilities and resilience skills, identity/ belonging and nationality might cause one to hypothesize differences in responses to the qualitative prompts.

The discussion raises some interesting points and would benefit from some deeper exploration. For example, the authors state that “Our findings can be interpreted as a readily accessible, pedagogically inclusive and interactive means for URM postdocs to advance their skills, thus reducing the mentoring gap.” Do the authors have evidence that the learners felt the modules filled in a gap in their mentoring? Is it possible that the course did not fill a gap but rather heightened awareness about the need to expand their mentoring network to meet needs outside the course? Another example: the authors state “In the third module, Developing Resilience, we engage postdocs in learning, reflecting on, and applying resilience skills, and provided more information about resources and support services for emotional support or mental and psychological well-being.” Do the authors have data from learners that these resources were useful?

Minor edits:

• I believe the reference to the S1 text should be to S2 on page 11

• The identification of the quotes from URM or non-URM should be consistent in the results.

6. PLOS authors have the option to publish the peer review history of their article (what does this mean?). If published, this will include your full peer review and any attached files.

Reviewer #1: No

Reviewer #2: **Yes: **Christine Pfund

---

## [Author Response · Author response to Decision Letter 0]

15 Apr 2023

We uploaded a response to the reviewer comments in the document section. Please let us know if further information is needed.

---

## [Decision Letter · Decision Letter 1]

9 May 2023

A national professional development program fills mentoring gaps for postdoctoral researchers

PONE-D-22-26426R1

Dear Dr. Hokanson,

We’re pleased to inform you that your manuscript has been judged scientifically suitable for publication and will be formally accepted for publication once it meets all outstanding technical requirements.

Kind regards,

Sue Ellen DeChenne-Peters, Ph.D.

Academic Editor

PLOS ONE

Additional Editor Comments (optional):

Reviewers' comments:

Reviewer's Responses to Questions

**Comments to the Author**

1. If the authors have adequately addressed your comments raised in a previous round of review and you feel that this manuscript is now acceptable for publication, you may indicate that here to bypass the “Comments to the Author” section, enter your conflict of interest statement in the “Confidential to Editor” section, and submit your "Accept" recommendation.

Reviewer #1: All comments have been addressed

Reviewer #2: All comments have been addressed

2. Is the manuscript technically sound, and do the data support the conclusions?

Reviewer #1: Yes

Reviewer #2: Yes

3. Has the statistical analysis been performed appropriately and rigorously? 

Reviewer #1: No

Reviewer #2: Yes

4. Have the authors made all data underlying the findings in their manuscript fully available?

Reviewer #1: Yes

Reviewer #2: Yes

5. Is the manuscript presented in an intelligible fashion and written in standard English?

Reviewer #1: Yes

Reviewer #2: Yes

6. Review Comments to the Author

Reviewer #1: This revised manuscript adequately addressed the reviewers concerns. It is a well-written manuscript that presents the impact of an online course, showing significant improvements in those who took the course.

Reviewer #2: (No Response)

7. PLOS authors have the option to publish the peer review history of their article (what does this mean?). If published, this will include your full peer review and any attached files.

Reviewer #1: No

Reviewer #2: **Yes: **Christine Pfund

---

## [Editor Report · Acceptance letter]

2 Jun 2023

PONE-D-22-26426R1 

A national professional development program fills mentoring gaps for postdoctoral researchers 

Dear Dr. Hokanson:

I'm pleased to inform you that your manuscript has been deemed suitable for publication in PLOS ONE. Congratulations! Your manuscript is now with our production department. 

Kind regards, 

on behalf of

Dr. Sue Ellen DeChenne-Peters 

Academic Editor

PLOS ONE